# An application of YOLOv8 integrated with attention mechanisms for detection of grape leaf black rot spots

**Jiajun Zhu**[1], **Jinlin Qiu**[2], **Shouwen Chen**[3], **Shuxi Chen**[1], **Haifei Zhang**[4]*

**1** College of Yonyou Digital Intelligence, Nantong Institute of Technology, Nantong, China, **2** College of Information Science and Technology, Nantong University, Nantong,China, **3** Institute of Mathematics and Finance, Chuzhou University, Chuzhou, China, **4** School of Information Engineering, Nantong Institute of Technology, Nantong, China

* 20229161@ntit.edu.cn

## Abstract

Aiming at the problems of low recognition rate of small target spots in grape leaf images and low detection accuracy due to low resolution of input images. In this paper, an improved recognition network based on YOLO v8 is constructed. In the constructed network, Spatial Pyramid Dilated Convolution (SPD-Conv) is used to replace each stepwise convolution layer and each pooling layer to better capture the detailed features of small targets. Meanwhile, the Efficient Multi-Scale Attention (EMA) Module is incorporated into the Neck part of YOLO v8 to make full use of the feature information of each detection layer and improve the accuracy of feature representation.The Plant Village dataset and the orchard image set are used to test the network performance of the improved model. The experimental test results show that the improved YOLO v8 has 92.64% precision, 93.28% recall and 96.17% AP. The size of model was a mere 7.1M. Compared to YOLO v8, the improvements are 2.38%, 1.91%, and 1.13%, respectively. Compared with the mainstream networks YOLO v4, YOLO v5, YOLO v6, and YOLO v7, precision is improved by 4.74%, 3.38%, 4.15%, and 4.69%, respectively. Therefore, the improved network proposed in this paper can improve the detection accuracy of small target objects and also identify the black rot disease of grape leaves more accurately.

## Introduction

In recent years, China's grape industry is in a critical period, with a steady increase in acreage and production, and a transition to a modern, high-quality industry. However, grapes are often interfered with by a variety of diseases during the growth process, which seriously impede the normal growth of grapes. It affects the yield and quality of grapes, and brings losses to the economy of fruit farmers as well. Nowadays, it is inefficient to detect the growth condition of grapes by manual means, and there is a certain labour cost. Therefore, timely and accurate detection of grape diseases is a hot demand in smart agriculture [1].

Traditional disease recognition methods for machine vision and image processing extract external features from colour [2], shape [3], texture [4] and wavelet features [5] or

**Data availability statement:** The datasets generated during and/or analyzed during the current study are available from the corresponding author on reasonable request. Information on the open dataset is available at the web site below：https://plantvillage.psu.edu/blogposts/323-trees-for-money-and-food-how-plantvillage-s-agroforestry-initiative-is-transforming-the-world,

**Funding:** Natural Science Foundation of Nantong City (JC2023075); 2)Social Livelihood Science and Technology Plan of Nantong City (MSZ2023175). 3)Jiangsu Provincial Natural Science Foundation for Higher Education Institutions (22KJB520032); The funders had no role in study design, data collection and analysis, decision to publish, or preparation of the manuscript.

**Competing interests:** The authors have declared that no competing interests exist.

combinations, and classify the extracted features using classifiers for crop disease classification. Majumdar et al. [6] proposed a combination of K-meas clustering and Artificial Neural Networks (ANN) for plant disease identification method. The K-meas clustering of the features of the disease and ANN classification of the features were used to achieve wheat disease recognition with 85% recognition accuracy. Pantazi et al. [7] proposed a method for disease feature extraction using Local Binary Patterns (LBPs). Different grape diseases were used for experimental testing and the results showed that the recognition accuracy of individual grape diseases reached 95%. Wang et al. [8] proposed a visual spectral based method for cucumber powdery mildew leaf recognition. The method extracted spectral features in the band 450–780 nm and then used SVM for classification, obtaining 98.13% recognition accuracy. Rahaman et al. [9] used k-means to segment leaf spots, and the extracted colour and texture features were input into KNN for classification, obtaining 96.76% recognition accuracy. Although traditional recognition methods can accomplish the identification and detection of diseases. However, the recognition accuracy of the existing algorithms still cannot meet the needs of agricultural production, and the complex feature extraction link needs to be further improved and refined.

With the development of artificial intelligence technology, convolutional neural network (CNN) is applied to the field of agriculture. Such as plant disease detection [10–11], plant species detection [12–13] and disease level assessment [14–15]. The application of these technologies makes agricultural production develop in the direction of more intelligent and efficient. In a study focused on grape leaf diseases, Wagh et al. [16] proposed an Alex-Net model based on convolutional neural networks for the classification of grape leaf diseases. Experimental results revealed that the recognition accuracy of grape leaf diseases reached 98.23%. Ji et al. [17] introduced a novel CNN framework, termed UnitedModel, designed to identify various grape leaf diseases. After training the network on four distinct types of leaf diseases and testing it on the Plant Village dataset, an accuracy of 98.57% was achieved. Liu et al. [18] proposed a DICNN model for the classification and identification of six diseases in grape leaves. The model achieved an identification accuracy of 97.22%, outpacing GoogLeNet and ResNet-34 by 2.97% and 2.55%, respectively. Chao et al. [19] enhanced the SE_Xception network by incorporating the SENet module, resulting in a lightweight model for identifying grape leaf diseases. This model outperformed Xception, MobileNetV1, and ShuffleNet by 1.99%, 1.60%, and 1.22%, respectively, in terms of identification accuracy. Wang et al. [20] built upon the ShuffleNet-v2 model by integrating ECA, constructing an improved ECA-SNet model for grape leaf disease identification. In comparative experiments, this model attained an accuracy of 98.86%. While significant progress has been made in the identification and classification of grape leaf diseases [21], accurately and efficiently locating the position of these diseases remains a challenge not addressed by the aforementioned methods.

Xie et al. [22] have developed an enhanced model for leaf disease detection, termed Faster DR-IACNN, which is capable of real-time monitoring of grape growth processes. By incorporating various attention modules, the model significantly enhances the detection of grape leaf diseases, achieving an accuracy rate of 81.1%. Tang et al. [23] proposed a lightweight method for grape disease detection that integrates channelwise attention (CA) and is based on the ShuffleNet architecture, enabling the detection of multiple disease targets within a single image. Their approach attained a detection accuracy of 99.19% across four types of grape diseases. Narasimman et al. [24] introduced a lightweight approach for grape leaf disease detection, leveraging depthwise separable convolutions to extract deep features from images. Additionally, their refined model minimizes computational costs and improves training outcomes. Tardif et al. [25] devised a method for detecting diseases in grape leaves by collecting and comparing diseased leaves from different grape varieties. Their experiments demonstrated

that the proposed method effectively diagnoses grape leaf diseases. Yeswanth et al. [26] proposed a grape leaf disease detection method based on a residual skip network, employing a collaborative loss function and guided filtering techniques to enhance the resolution of disease targets. The experiments under different super-resolution scaling factors showed promising results. By constructing convolutional neural network models, automatic identification and detection of grape leaf diseases can be achieved. However, during the detection of grape leaf diseases, there are challenges with small target lesions. The accuracy of detecting such small lesions remains to be improved. Moreover, the refined models result in increased computational and storage requirements, thus raising the operational costs.

Therefore, this article builds upon the foundational research conducted by our research group [27–28]. While meticulously balancing detection accuracy with computational model efficiency, this paper proposes a method for detecting black rot disease on grapevine leaves by fusing an attention module and a lightweight convolutional layer. Efficient Multi-Scale Attention (EMA) Module ensures the effective use of feature information at each detection level, which improves the accuracy of feature representation.Spatial Pyramid Dilated Convolution (SPD-Conv), replaces the original convolutional layer with a step size of 2 with a block constructed using SPD-Conv. This modification captures the details of small to medium sized targets more efficiently.

The main work of this paper is as follows:

(1) SPD-Conv is employed to substitute each stride convolution layer and each pooling layer, thereby capturing multi-scale features.

(2) The Neck part of YOLO v8 is incorporated into the EMA Module to enhance small target representation.

(3) Experiments are conducted to compare and analyse the effect of YOLO series algorithms with the algorithms proposed in this paper in terms of detection performance.

Shortcomings of the current study and future research directions:

(1) At present, the refined model is confined to the domain of detecting a single plant disease. Given the high cost associated with manual annotation, subsequent research will focus on testing the robustness of the model.

(2) During the initial phase of optimizing the model, factors affecting plant diseases in outdoor environments were not considered. However, the improved model has demonstrated enhanced performance compared to the baseline model. Future research will prioritize the analysis of plant diseases in outdoor settings, continuously refining the plant disease detection model to provide technical support for the development of mobile products.

## Materials and methods

### Image dataset

The experimental data for this paper were obtained from open datasets-Plant Village [29] and Grape Orchard. Among them, 1180 images of grapevine black rot from Plant Village were used for network training and model testing. Among them, 1,072 disease images were employed for the training of the model. Two test sets, labeled as test1 and test2, were separately utilized to assess the model's performance. Specifically, test1 comprised 108 disease images from Plant Village, while test2 included 108 images of grape black rot from an orchard, both sets being employed to evaluate the model. LabelImg software was used to label black rot target areas prior to testing. The total number of black rot spots labelled was approximately

17643, with an average of 16 target areas present in an image. The images of the experimental dataset are shown in Fig 1. In this case, Panel A is an image of black rot on grape leaves in a laboratory setting and Panel B is an image of black rot disease in an orchard setting.

## Grape leaf spot detection based on improved YOLOv8

**Efficient multi-scale attention.** In this paper, we introduce Efficient Multi-Scale Attention (EMA) [30], cross-space learning as well as non-dimensionality reduction of channels, which can improve feature processing for small targets. EMA dynamically adjusts the weights in the feature map in an adaptive manner according to the importance of each region, allowing the network to focus its attention on the small target part. The EMA module reshapes some of the channels into batch dimensions and divides the features into multiple groups for convolution in the channel dimensions, which reduces the amount of computational effort; spatial semantic features are uniformly distributed in each group of features, which avoids the impact of channel dimensionality reduction on the detection accuracy. The EMA structure is shown in Fig 2.

From the structure diagram, it can be seen that the EMA consists of three branching parts. The packet feature maps are extracted through two parallel paths in the 1 × 1 branch and one path in the 3 × 3 branch. In the 1 × 1 branch, two 1D global average poolings are used for

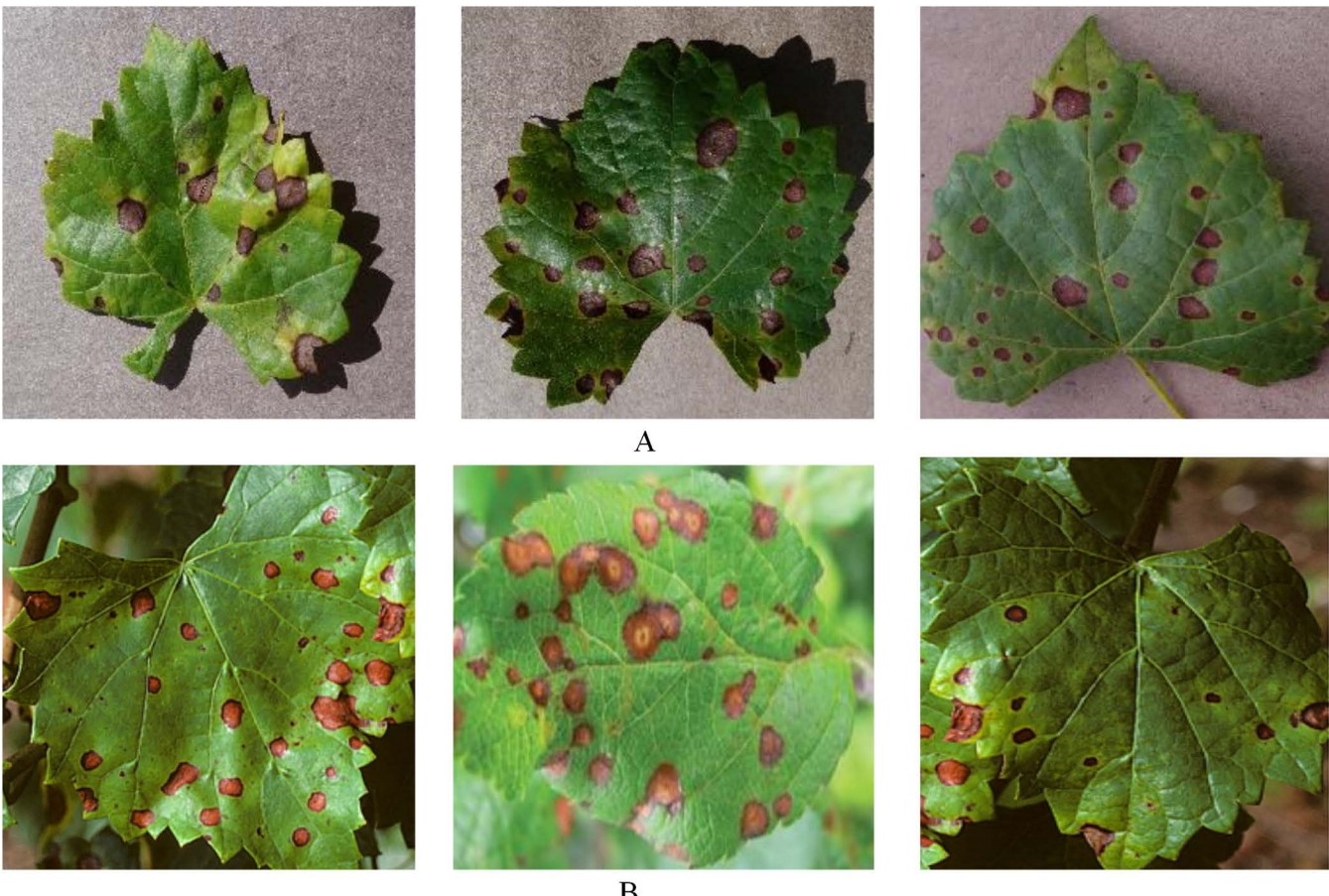

A

B

**Fig 1. Dataset: (A) The image of plant village; (B) The image of Orchard.**

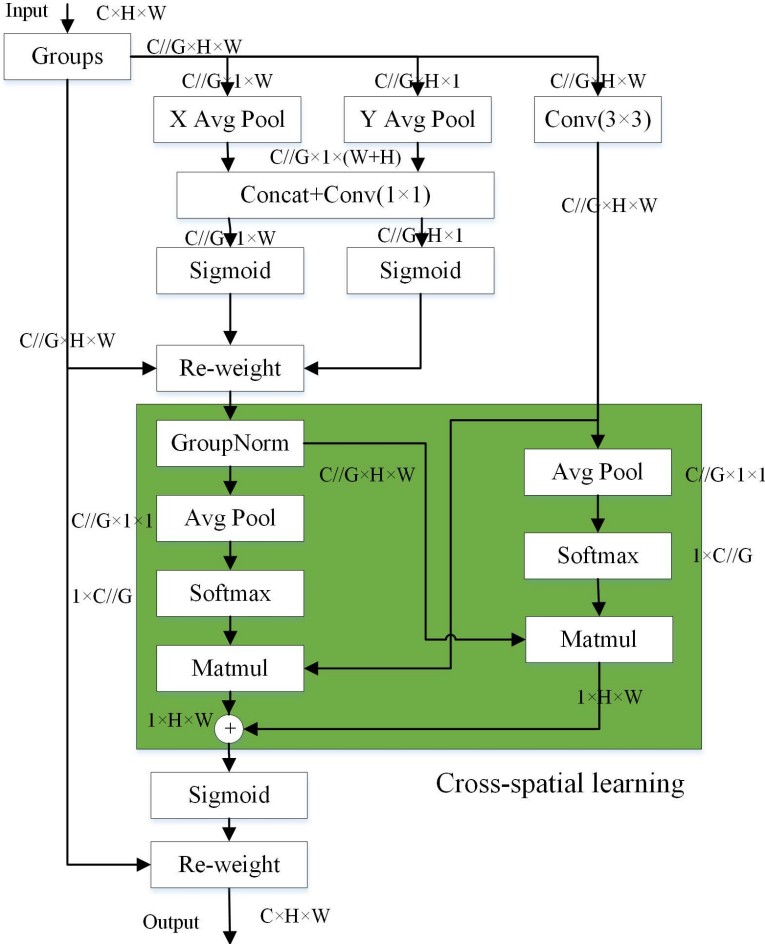

**Fig 2. Efficient multi-scale attention.**

channel coding in vertical and horizontal directions, respectively. Channel coding involves global feature extraction for each channel of the feature map through global average pooling, aiming to capture the global contextual information across channels. By performing 1D global average pooling separately in the vertical and horizontal directions, we can obtain global features for each channel in different orientations. These features contribute to enhancing the expressive and interpretative capabilities of the model. The pooling kernel sizes are of size (1,W) and (H,1) respectively.Multi-scale features are captured in the 3 × 3 branch using a 3 × 3 convolution. The two coded features are then connected along the image height direction using 2D global pooling, computed as shown in (1):

$$Z_c = \frac{1}{H \times W} \sum_{j}^{H}\sum_{i}^{W} x_c (i, j) \tag{1}$$

After 2D global mean pooling, the nonlinear activation function of Softmax was used to adapt the linear transformation. The Softmax function stands as a commonplace normalization tool within the sphere of deep learning. It metamorphoses a vector of real numbers into a probability distribution, ensuring that the resultant output for every element is confined within the spectrum of 0–1, and that the aggregate sum of these elements equals unity. Typically, the

Softmax function is employed in the output layer of neural networks, thereby eliciting the predicted probabilities for each class, which in turn clarifies the distribution of these predictive class probabilities. By multiplying the output of the parallel processing with the output of Softmax, a spatial attention map is obtained, fusing spatial information from different scales at the same stage. Similarly, another spatial attention map was generated using 2D global average pooling and Softmax's nonlinear activation function for features extracted from an ordinary convolution with a convolution kernel size of $3 \times 3$. The information of the generated spatial attention weight values is aggregated and finally the Sigmoid nonlinear activation function is used. The Sigmoid function, a widely employed nonlinear activation function in neural networks, adeptly compresses any real number into the interval (0, 1), making it particularly apt for binary classification scenarios where probabilistic output is required.

**Spatial pyramid dilated convolution.** To address the problem that the use of stepwise convolution and pooling for low-resolution images and small targets in the target detection task leads to the loss of important information and thus fails to extract the effective features in a better way. Sunkara et al. [31] proposed a Spatial Pyramid Dilated Convolution (SPD-Conv) module. SPD-Conv consists of one SPD (Space to depth) layer and 1 Non-strided convolutional layer, which is a technique to convert image spatial information into depth information, thereby reducing information loss and improving the network's ability to learn from low-resolution images and small targets. Fig 3 shows the schematic diagram of the SPD-Conv network when the scale factor (scale) is equal to 2. For any given feature map X with shape S×S×C1, four feature maps with shape (S/2,S/2,C1) are obtained through the SPD layer, and these sub-feature maps are connected along the channel dimension to obtain a feature map X' with shape (S/2,S/2,4C1). In order to preserve as much feature information as possible, the feature map X' is finally transformed into a feature map of shape (S/2,S/2,C2) using a convolution with a filter number of C2 and a step size of 1.

The grape leaf images in this paper are low-resolution images, while there are small disease targets in the images. Therefore, the original convolutional layer with a step size of 2 in YOLOv8 is replaced with a block constructed by SPD-Conv. It can capture the details of small and medium-sized targets more effectively and improve the model's ability to process low-resolution images and too small disease spots.

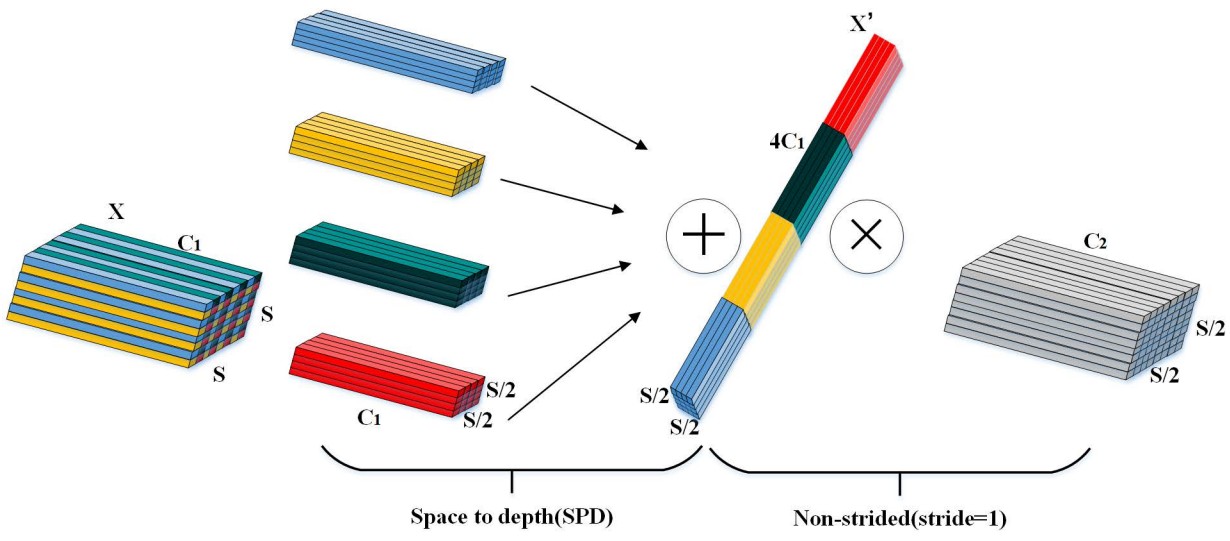

**Fig 3. Spatial pyramid dilated convolution (scale = 2).**

**Improved YOLOv8.** In this paper, a network model based on YOLOv8 is chosen as the final model for grape leaf black rot detection.The YOLO series [32] algorithm was proposed in 2016, which is a target detection algorithm that directly outputs the target category and probability without extracting the candidate region. Target detection models typically rely on Convolutional Neural Networks (CNNs) to extract feature maps from input images, which are subsequently employed for classification (identifying object types) and regression (localizing objects) tasks. In the context of detecting black rot in grapevine leaves, these feature maps need to capture multi-scale features, high-level semantic information, and spatial information. In this paper, the improved YOLOv8 consists of Backbone, Neck and Head. Its network structure is shown in Fig 4.

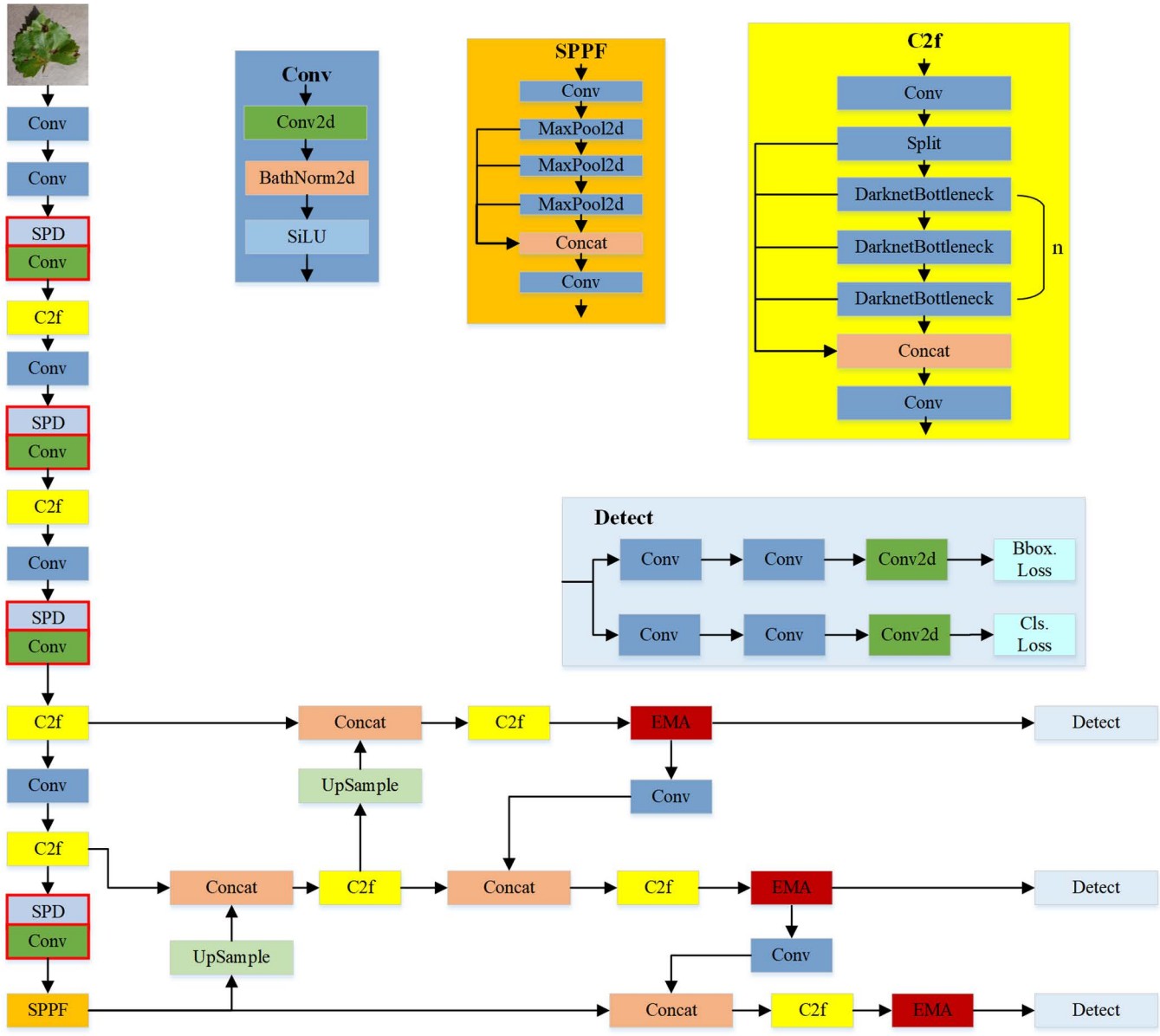

**Fig 4. Improved YOLOv8 network structure.**

The function of Backbone is to process the input image to generate the feature maps needed for target detection. Backbone consists of four modules, Conv, SPD-Conv, C2f, and SPPF. The Conv module consists of a convolutional layer, a batch normalisation, and an activation function, which performs convolutional operations to extract the local features of the image. During the image input stage, an input image of dimensions 640 * 640 * 3 undergoes processing through the Conv module, resulting in an output of dimensions 320 * 320 * 64.

The SPD-Conv, which is used to adjust the convolutional step size to reduce information loss and improve the network's learning ability for low-resolution images and small targets. The SPD-Conv module incorporates both SPD and Conv layers; the SPD layer rearranges feature maps through downsampling operations, transforming spatial information into higher dimensions. Specifically, it converts an input image of dimensions 320 * 320 * 64 into a 160 * 160 * 256 image by scaling spatial dimensions to depth with a factor of 2. Unlike traditional strided convolutions, the non-strided convolutional layer does not skip pixels as its kernel slides over the feature map; instead, it moves with a stride of 1. This operation processes the 160 * 160 * 256 image, producing an output of 160 * 160 * 128. This approach ensures that crucial feature information can be extracted from the original feature maps in subsequent tasks, making it particularly effective for detecting grape leaf diseases in scenarios with small targets and low resolution.

The function of the C2f module is to increase the network depth and sensory field to enhance the ability of feature extraction. Upon undergoing the C2f transformation, the input and output images maintain their dimensions unaltered, specifically at 160 * 160 * 128. SPPF is the spatial pyramid pooling module, which has the function of pooling the features at different scales to obtain multi-scale information. The essence of this module is that after a convolutional layer, with the different scales to do the maximum pooling operation, and then splicing the features to fuse the different scale features.

The Neck part of YOLOv8 adopts the FPN+PAN structure, which further processes the backbone network features and fuses the features from different layers to improve the detection performance. Due to the downsampling operation of YOLOv8 algorithm, the feature information of small targets is gradually reduced, which causes the algorithm to reduce the feature processing capability of small targets, and is easy to cause misdetection and omission detection of small targets. Therefore, as shown in the red module in the figure, the EMA module is incorporated after the C2f in the Neck part. The Head part adopts Decoupled-Head, which performs the classification task and the detection task separately. Decoupled-Head has the function of being able to separate the feature extraction and the pixel prediction, which allows the network to better deal with the different feature scales and semantic information, and also better solves the problem of different focuses of classification and localisation.

**Evaluation metrics.** In order to evaluate the network performance of the improved model, precision (P), recall (R) and average precision (AP) were used as the evaluation metrics in this paper. p denotes the number of correctly detected data/total number of detections. r denotes the number of correctly detected data/number of all the positive data in the ground truth. ap serves as the overall performance measure, considering P-R trade-offs. overall performance measure, considering P-R trade-offs. The evaluation metrics are calculated as follows:

$$P = \frac{TP}{TP + FP} \times 100\% \tag{2}$$

$$R = \frac{TP}{TP + FN} \times 100\% \tag{3}$$

$$AP = \int_0^1 P(R)dR \times 100\% \tag{4}$$

where TP denotes true positives, i.e., correctly detected black rot spots. FP denotes false positives, i.e., the model incorrectly detects the background as black rot spots. FN denotes false negatives, i.e., black rot spots not detected by the model.

The SPD-Conv, which is used to adjust the convolutional step size to reduce information loss and improve the network's learning ability for low-resolution images and small targets. The SPD-Conv module incorporates both SPD and Conv layers; the SPD layer rearranges feature maps through downsampling operations, transforming spatial information into higher dimensions. Specifically, it converts an input image of dimensions 320 * 320 * 64 into a 160 * 160 * 256 image by scaling spatial dimensions to depth with a factor of 2. Unlike traditional strided convolutions, the non-strided convolutional layer does not skip pixels as its kernel slides over the feature map; instead, it moves with a stride of 1. This operation processes the 160 * 160 * 256 image, producing an output of 160 * 160 * 128. This approach ensures that crucial feature information can be extracted from the original feature maps in subsequent tasks, making it particularly effective for detecting grape leaf diseases in scenarios with small targets and low resolution.

The function of the C2f module is to increase the network depth and sensory field to enhance the ability of feature extraction. Upon undergoing the C2f transformation, the input and output images maintain their dimensions unaltered, specifically at 160 * 160 * 128. SPPF is the spatial pyramid pooling module, which has the function of pooling the features at different scales to obtain multi-scale information. The essence of this module is that after a convolutional layer, with the different scales to do the maximum pooling operation, and then splicing the features to fuse the different scale features.

## Results

### Experimental platform and network parameters

The testbed of this paper is based on a desktop computer under Windows 10 and relies on Anconda software. PyTorch 1.8.0 was used as the deep learning framework required for training. CUDA version is 10.2, and python version is 3.9.0. the GPU model is NVIDIA GeForce GTX 1080ti, and an AMD Ryzen 5 1600X Six-Core Processor. the improved The network parameters of YOLOv8 are configured as shown in Table 1 below.

### Training results of YOLOv8 network

In this paper, 1070 grape leaf black rot images containing 17643 target regions after labelling were input into the YOLOv8 network for training. The training time of the network is 7.35h,

Table 1. Network parameter configuration.

| Network parameter | Set value |
|---|---|
| Epoch | 250 |
| Batch | 3 |
| Image size | 640 * 640 |
| Learing rate | 0.01 |
| Momentum | 0.937 |
| Training ratio | 9:01 |
| IOU | 0.7 |

and the training result data is shown in Fig 5. The training result data contains information from both the training part and the validation part of the network. The results show that the main model evaluation indexes P is 90.26%, R is 91.37% and mAP is 95.34%. In this paper, there is only one type of detection target, black rot disease. So, the evaluation metric mAP=AP in this paper.

### Improved training results of YOLOv8 network

Similarly, the labelled 1070 images of grape leaf black rot containing 17643 target regions were fed into the improved YOLOv8 network for training. The training time of the network was 6.32h, and the training result data is shown in Fig 6. The training result data contains information from both the training part and the validation part of the network. The results show that the main model evaluation metrics P is 92.64%, R is 93.28% and mAP is 96.17%. Compared to the network training results of YOLOv8, the network training results of improved YOLOv8 are smoother.

## Discussion

### Previous research

In comparison to the content of the Group's earlier research, entitled "Grape Leaf Black Rot Detection Based on Super-Resolution Image Enhancement and Deep Learning," the YOLO v3 model was employed as the tool for detecting black rot in grape leaves. Prior to finalizing

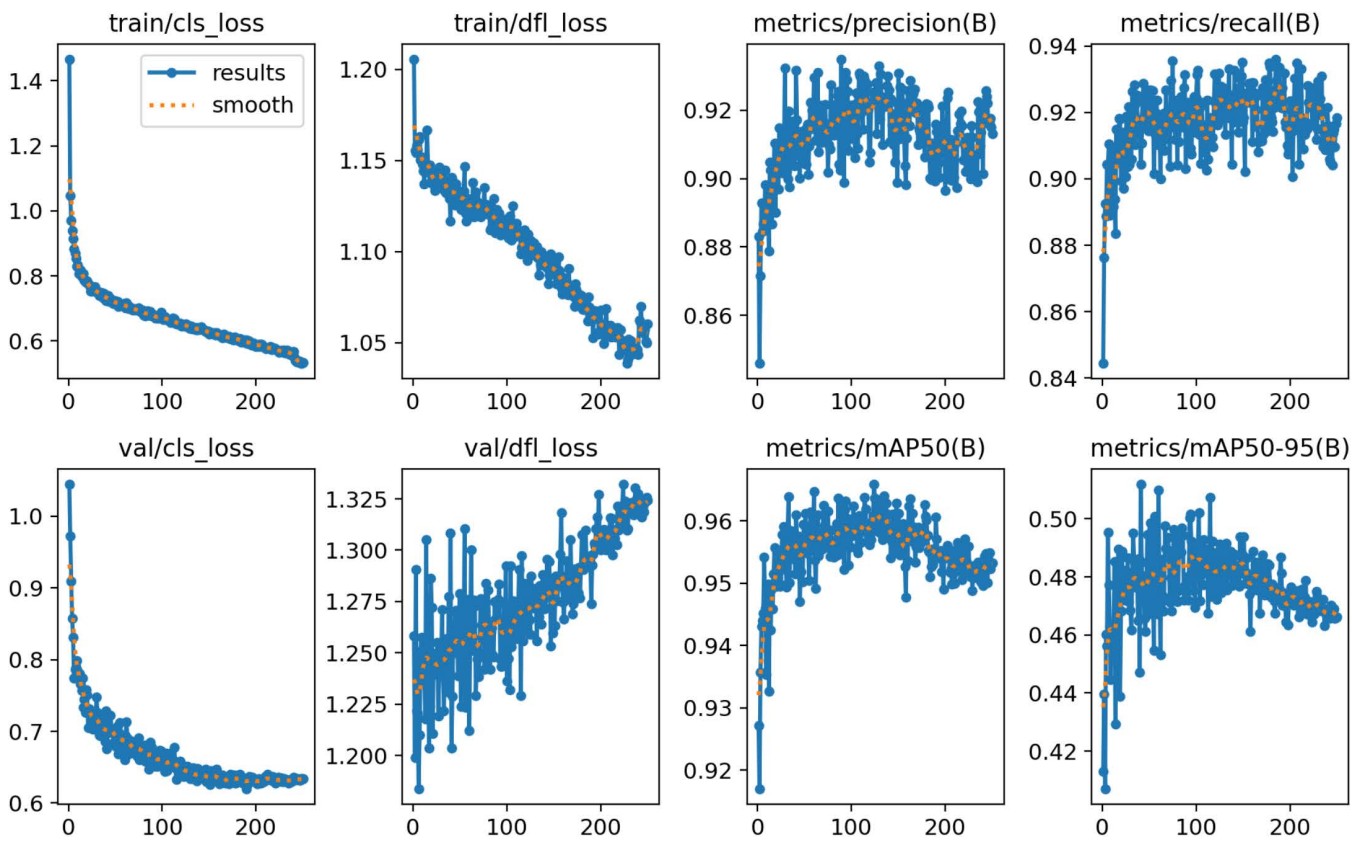

**Fig 5. Training results of YOLOv8 network.**

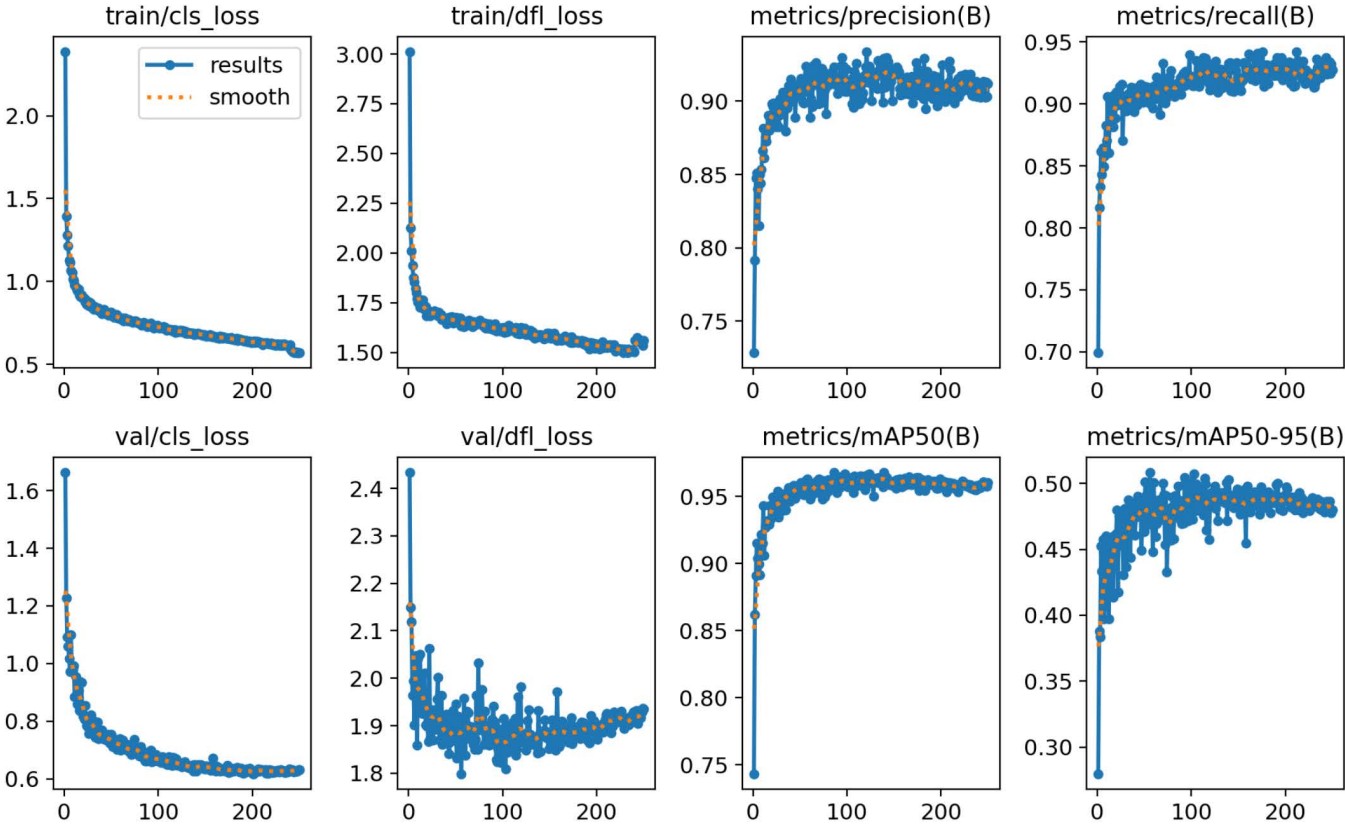

**Fig 6. Training results of improved YOLOv8 network.**

YOLOv3 as the definitive disease detection model, a comparative evaluation was conducted among a series of detection algorithms, including Faster-RCNN, SSD, and RCNN. Ultimately, YOLOv3 was selected based on its superior model evaluation metrics, P and R, which were 89.85% and 83.75%, respectively. Despite these metrics, there remains substantial room for improvement. As illustrated in Fig 7, the detection results are based on the YOLOv3 model for black rot lesion detection. It is evident from the results that small target lesions were not detected. In response to such instances where small target diseases were undetected by the algorithm, the Group has continuously engaged in research and refinement. This endeavor culminated with the introduction of the latest version, YOLO v8, which has now been adopted as the Group's most recent model for detecting black rot in grape leaves. Before incorporating YOLO v8 as the latest model for black rot disease detection, extensive comparisons were made within the YOLO series of algorithms.

## Comparison with other methods

In order to test the improved YOLOv8 model for the detection of grape leaf disease images. Also, to confirm the ultra-high detection performance of YOLOv8 model itself. Some advanced models were chosen as comparisons in this experiment. Including YOLOv4, YOLOv5, YOLOv6 and YOLOv7, which are YOLO series models. For different detection models, this experiment keeps all kinds of parameters mentioned above unchanged during network training. The recognition results of different detection models are shown in Table 2.

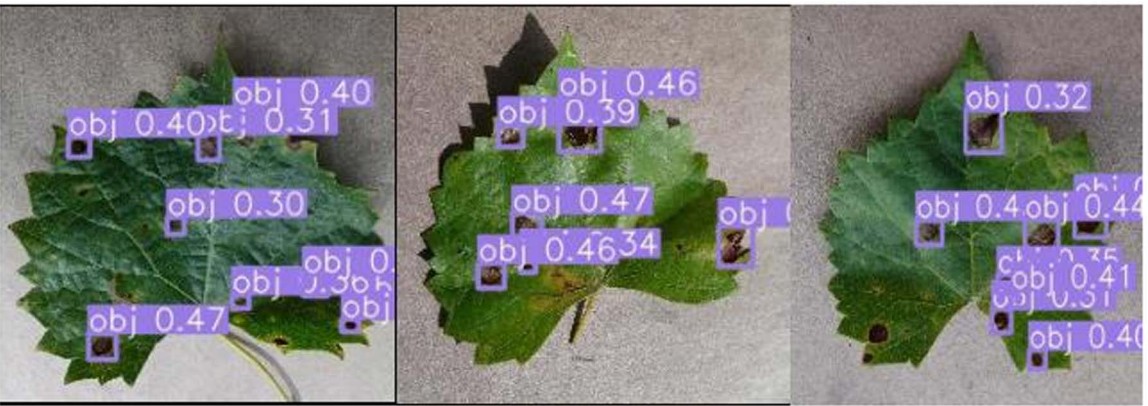

**Fig 7. Detection results of YOLO v3.**

From the table, it can be seen that different versions of YOLO models showed good detection of black rot. However, YOLOv8 is higher than the other four models in P, R and AP indexes. Meanwhile, YOLOv8 is an improved and optimised model based on YOLOv5, and its network performance is theoretically better than that of YOLOv5. Compared with the other models, the network performance of YOLOv5 model is also better than that of the other versions. For the grape leaf black rot studied in this paper, the network performance of the improved YOLOv8 is higher compared to the different models compared in this paper. On the evaluation metric P, the improved YOLOv8 is higher than the other models by 4.74%, 3.38%, 4.15%, 4.69% and 2.38%, respectively. On evaluation metric R, improved YOLOv8 is higher than other models by 7.55%, 4.82%, 6.96%, 8.12% and 1.91%, respectively. Similarly, on AP, the improved YOLOv8 model is higher than other models. Therefore, the improved YOLOv8 model proposed in this paper can be more effective in detecting black rot disease. Meanwhile, the proposed improved model can also solve the problem of small target detection.

This paper uses an improved model for data testing. The precision-recall (P-R) curve of our proposed method on the test set is shown in Fig 8. In the figure, "b" denotes the detection target Black Rot, with an AP (Average Precision) value of 96.17%.

## Ablation experiment

In order to verify the effectiveness of SPD-Conv and EMA attention mechanism on the algorithm of this paper, the design of ablation experiments is carried out in this paper using YOLOv8 as a benchmark combined with the improvement module. The experimental results are shown in Table 3. From Table 3, it can be seen that after adding SPD-Conv alone in the

**Table 2. Statistical analysis of recognition results by different detection models.**

| Model | Precision (%) | Recall (%) | AP (%) |
| --- | --- | --- | --- |
| YOLOv4 | 87.90 | 85.73 | 93.65 |
| YOLOv5 | 89.26 | 88.46 | 94.67 |
| YOLOv6 | 88.49 | 86.32 | 94.50 |
| YOLOv7 | 87.95 | 85.16 | 94.36 |
| YOLOv8 | 90.26 | 91.37 | 95.34 |
| Ours | 92.64 | 93.28 | 96.17 |

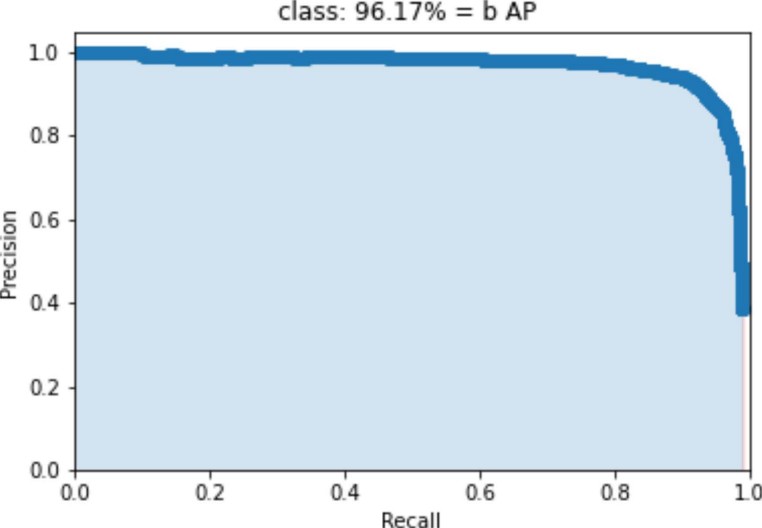

**Fig 8. P-R curves.**

**Table 3. Statistical table of ablation experiments.**

| Experiment | SPD-Conv | EMA | P(%) | R(%) | AP(%) | PT |
|---|---|---|---|---|---|---|
| 1 | | | 90.26 | 91.37 | 95.34 | 6.1M |
| 2 | √ | | 92.86 | 92.12 | 95.73 | 7.14M |
| 3 | | √ | 91.47 | 93.65 | 96.26 | 6.12M |
| 4 | √ | √ | 92.64 | 93.28 | 96.17 | 7.16M |

YOLOv8 network,the method improves the detection effect of the model on low-resolution images and small targets. Compared with the YOLOv8 model, the detection accuracy P is significantly improved, and the P value is improved by 2.6%.

In terms of the PT weight files, the inclusion of SPD-Conv independently in the YOLOv8 model has resulted in an increase in the parameter count. Upon analysis, it has been considered that SPD-Conv generates multi-scale feature maps by performing pooling at varying scales. Consequently, the computational burden might escalate, as it necessitates pooling operations on feature maps of disparate scales. The feature fetching ability of the model is improved by adding the EMA attention mechanism alone in the YOLOv8 network. There is a significant improvement in P, R and AP values, where the value of R is improved by 2.28% and P by 1.21%. Considering the different gains that SPD-Conv and EMA each bring to the model, in this paper, both are added together to the YOLOv8 network. The P of the model rises to 92.64%, a 2.38% improvement over the original YOLOv8 algorithm. The model R reaches 93.28%, an improvement of 1.91% compared to the original algorithm. The size of the PT file has also increased by approximately 1MB due to the incorporation of SPD-Conv. Given that the improvement in detection accuracy has been achieved, the addition of approximately 1MB to the PT file size is theoretically acceptable in the context of mobile development.

Fig 9 shows the comparison of P, R indexes of this paper's algorithm under different improvements. From the figure, it can be seen that the YOLOv8 model has large fluctuations in the values of P and R during the network training process and lacks stability. The final average value on the evaluation index P is 90.26%, and the final average value on the evaluation

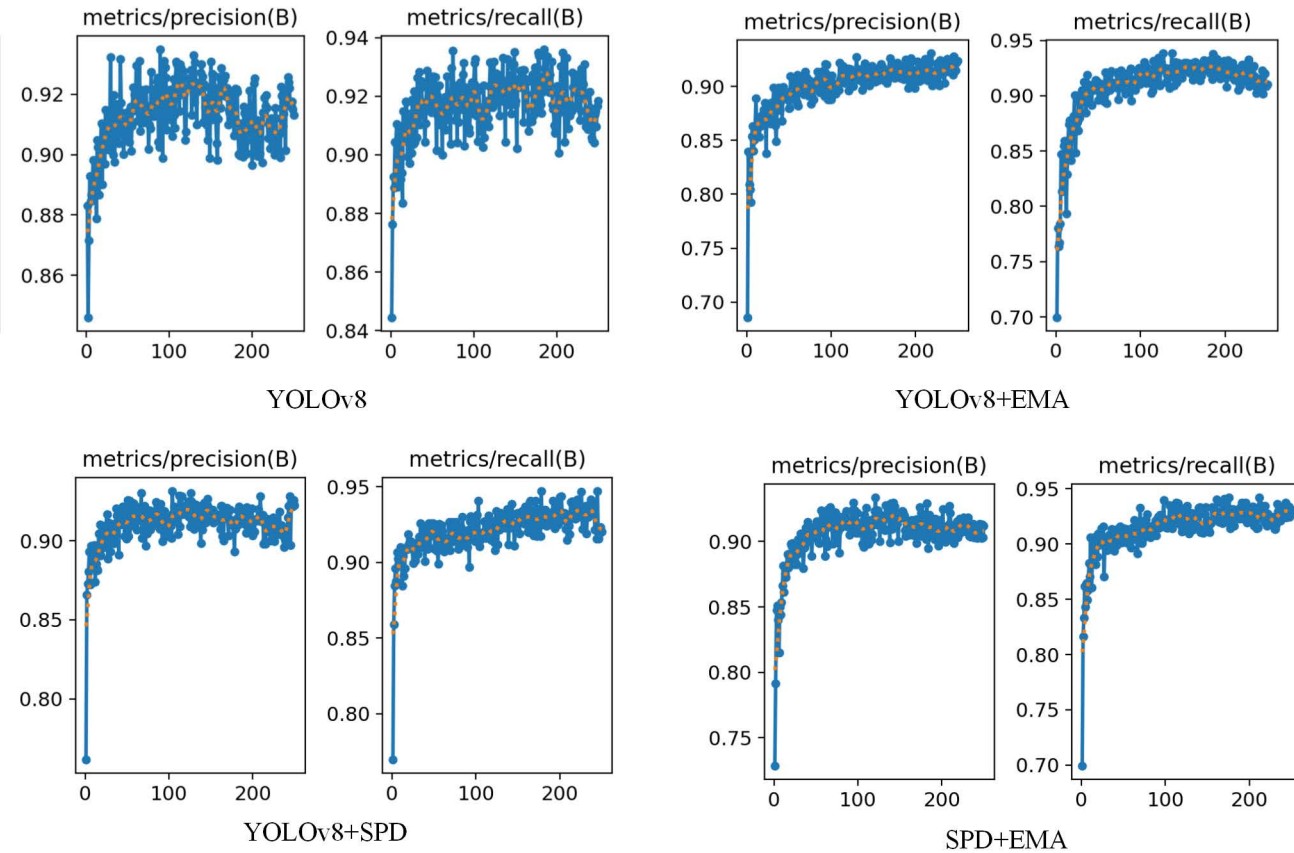

**Fig 9. Comparison of precision and recall metrics for different models.**

index R is 91.37%. Compared to the YOLOv8 model, the network performance is improved with the addition of EMA. As can be seen from the figure, the values of P and R gradually smoothed and converged to stability as the number of network trainings increased. When the network training reaches 100 epoch, the model tends to stabilize. In the evaluation index R, the YOLOv8 model after adding EMA consists of a significant improvement, and the final value reaches 93.65%.

Compared to YOLOv8, after replacing it with SPD, the P value is improved and the values of P and R are stabilized. As can be seen from the figure, the model is trained to 20 epoch, the network curve converges and the model is stabilized. On the evaluation metric P, the YOLOv8 model after incorporating SPD-Conv is significantly improved and the final value reaches 92.86%. Compared with these three types of networks, YOLOv8 after incorporating EMA and SPD-Conv is improved in both P and R values, which is compatible with the advantages of the two improvements. At the same time, the P and R values during the training of the network become more stable with the increase in the number of trainings. As can be seen from the figure, the model reaches stability at 60 epochs of training and the curve shows convergence. The YOLOv8 after fusion of EMA and SPD-Conv is finally stabilized at 92.64% and 93.28% in terms of P, R values. In summary, the improved YOLOv8 model proposed in this paper has higher detection effect and can better detect small target objects.

## Detection effect of the improved YOLOv8

In this paper, black rot disease detection is performed for the improved YOLOv8 model. A total of 2 test sets were used to validate the performance of the improved YOLOv8 model. The Test1 dataset contains 108 image data of grapes from Plant Village. The Test2 dataset contains 108 image data of grapes from an orchard environment. The results of the Test1 detection are shown in Fig 10, with a1-a4 being the results of the YOLOv8 detection, and b1–b4 are the improved YOLOv8 detection results. In the figure, the yellow circles indicate the spots that YOLOv8 failed to detect and the improved YOLOv8 was able to detect. A1–a4 Figures, each of them is marked with multiple spots, and multiple spots are small target spots. As can be seen from the figures, the improved YOLOv8 is better at detecting small target spots. Meanwhile, the detection performance is also more accurate and comprehensive, which also provides technical support for later mobile deployment.

In this paper, in order to verify the robustness and generalisation of the improved model. The Test2 dataset is used to test the model performance. The test results are shown in Fig 11, c1-c4 are the YOLOv8 detection results and d1-d4 are the improved YOLOv8 detection results. As can be seen from the figure, the YOLOv8 detection images are marked by purple circles. Areas of diseased spots existed where they were marked by purple circles, but were not detected correctly. As shown in the figure, the improved YOLOv8 provides highly accurate detection even in an orchard environment. As can be seen from the marked purple circles, the improved YOLOv8 is more capable of detecting small spots, and the incorporation of SPD-Conv and EMA enhances the detection of small spots in the YOLOv8 model. Compared to the YOLOv8 model, the improved YOLOv8 model is also more comprehensive in

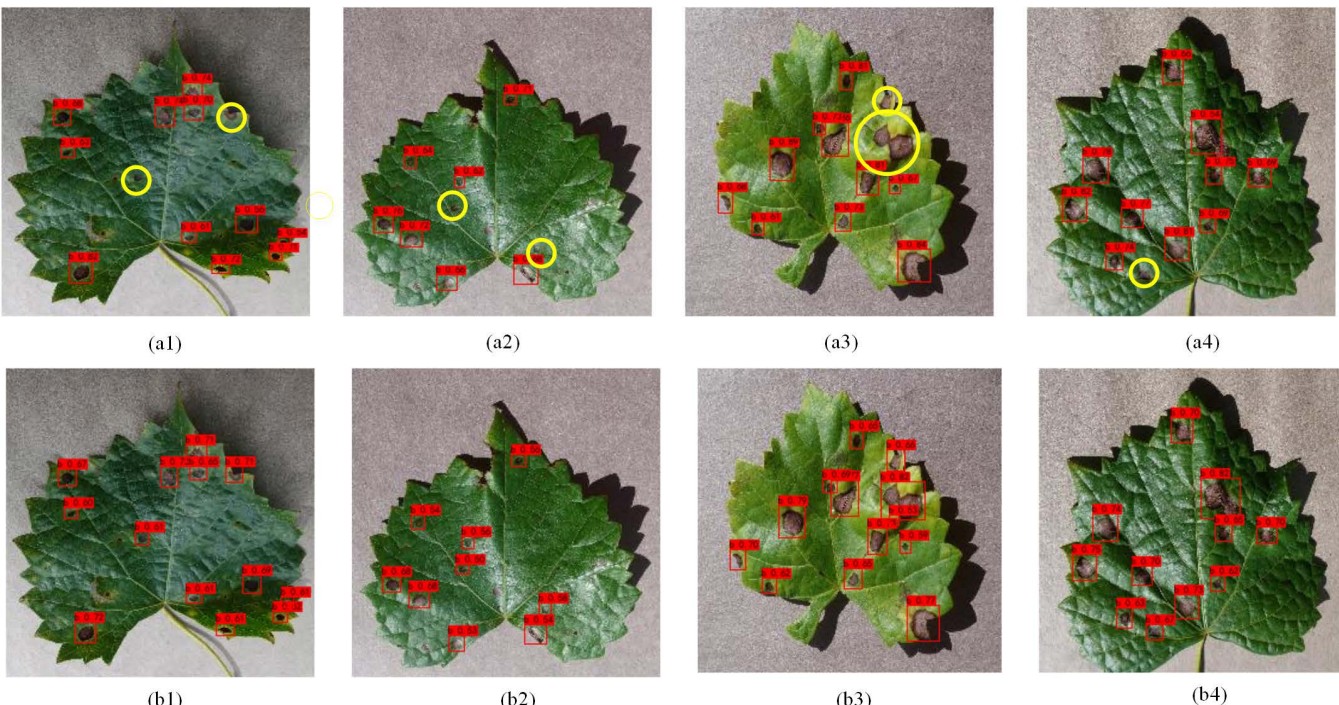

**Fig 10. Comparison of Detection Results for Test1: (a1–a4) are the detection results of the YOLOv8 model; (b1–b4) are the detection results of the imprvoed YOLOv8 model.**

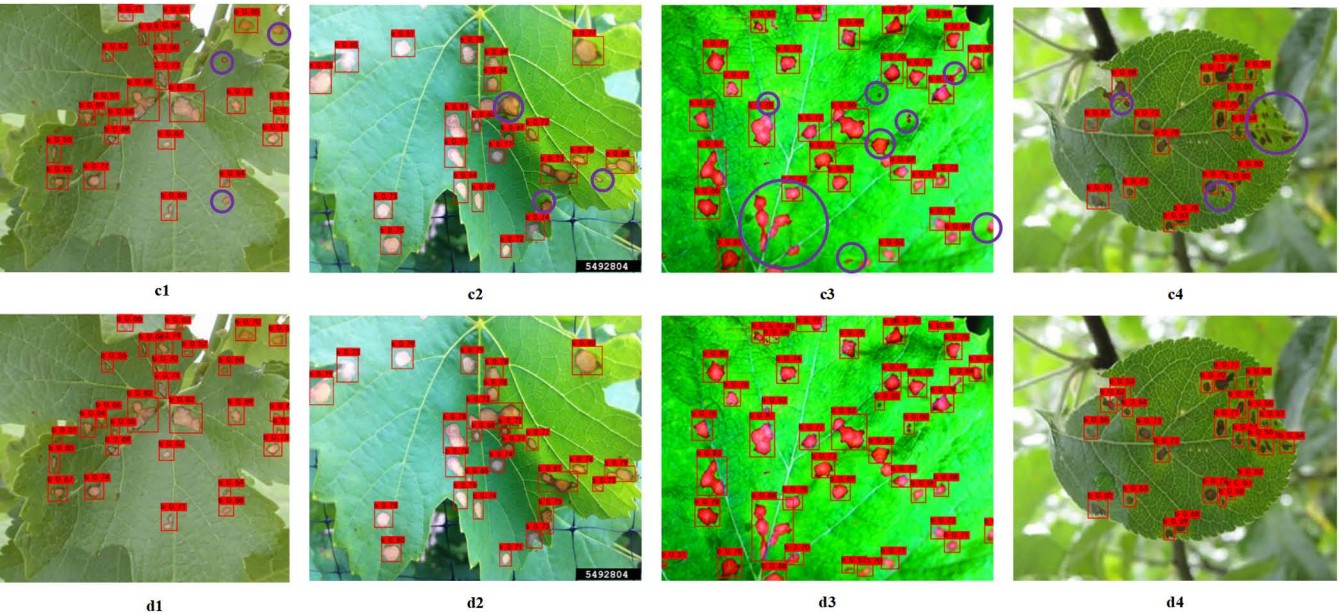

**Fig 11. Comparison of Detection Results for Test2. (c1–c4) are the detection results of the YOLOv8 model; (d1–d4) are the detection results of the imprvoed YOLOv8 model.**

detecting black rot spots and has a higher model recall. In summary, the improved YOLOv8 model proposed in this paper can more accurately detect black rot spots on grape leaves.

The detection statistics for Test1 and Test2 test sets are shown in Table 4. From the table, it can be seen that there are a total of 1532 diseased spots in the Test1 test set and a total of 1272 diseased spots in the Test2 test set. For the test results of Test1 dataset, YOLOv8 detected a total of 1,399 correct spots, and the improved YOLOv8 detected a total of 1,429 correct spots. In terms of the number of TPs and TNs, 103 spots were not detected in improved YOLOv8 and 133 spots were not detected in YOLOv8. The improved YOLOv8 model detected 30 more spots, and as can be seen from the detection plots, most of the spot types were small target objects. On FP, the improved YOLOv8 incorrectly detected 113 spots and YOLOv8 incorrectly detected 151 spots, and the improved YOLOv8 incorrectly detected 38 fewer spots than YOLOv8. From the detection results of Test1 dataset in the table, it can be seen that the improved YOLOv8 model has better detection results. For the test results of Test2 dataset, YOLOv8 detected a total of 1086 correct spots, and the improved YOLOv8 detected a total of 1134 correct spots. In terms of the number of TPs and TNs, 138 spots were not detected in improved YOLOv8 and 186 spots were not detected in YOLOv8. The improved YOLOv8 model detected 48 more spots, dominated by missed detection of small target spots. On FP, the improved YOLOv8 incorrectly detected 190 spots and YOLOv8 incorrectly detected 218

**Table 4. Test result statistics of Test1 and Test2.**

| | | Objective | TP | FP | TN | P | R |
|---|---|---|---|---|---|---|---|
| **Test1** | YOLOv8 | 1532 | 1399 | 151 | 133 | 90.26 | 91.37 |
| | Ours | | 1429 | 113 | 103 | 92.64 | 93.28 |
| **Test2** | YOLOv8 | 1272 | 1086 | 218 | 186 | 83.24 | 85.43 |
| | Ours | | 1134 | 190 | 138 | 85.63 | 89.15 |

spots, and the improved YOLOv8 incorrectly detected 28 fewer spots than YOLOv8. From the detection results of the Test2 dataset in the table, it can be seen that the improved YOLOv8 model can effectively detect the black rot disease of grape leaves in the orchard environment. However, the accuracy of the improved YOLOv8 model in the orchard environment still needs to be improved in terms of the model evaluation indexes P and R. In the future, the research on grapevine diseases in orchard environment will also become the direction and focus of this paper.

## Analysis of test results in natural environment

Test2 was employed to evaluate the disease detection performance of the improved model, with data from Test2 sourced from a natural environment. As seen from Table 4 in Section 4.4, the performance metrics P and R for the Test2 dataset have decreased. The image data in Test1 originated from the Plant Village dataset, which features comparatively simple background environments. These images include those of single leaves without overlapping, free from obstructive backgrounds, and were captured under ideal indoor lighting conditions. In contrast, the images in Test2 encompass single leaf images, multiple leaf images, as well as images with obstructive backgrounds. Given the complexity of image acquisition in natural environments, this study conducted a classification analysis of disease images within Test2.

As illustrated in Fig 12, Column a represents the detection types for single leaf images, Column b for multiple leaf images, Column c for images without obstacles, and Column d for images with obstructive backgrounds. From the images in Column a, it is evident that even single leaf images are subject to interference from obstructive backgrounds. The images d1 and d2 in Column d demonstrate this clearly; d1's image was captured with background blurring, whereas d2 was not. The result of background blurring led to the black trunk in d1 not being incorrectly detected as a disease, while the black trunk in d2 was mistakenly identified

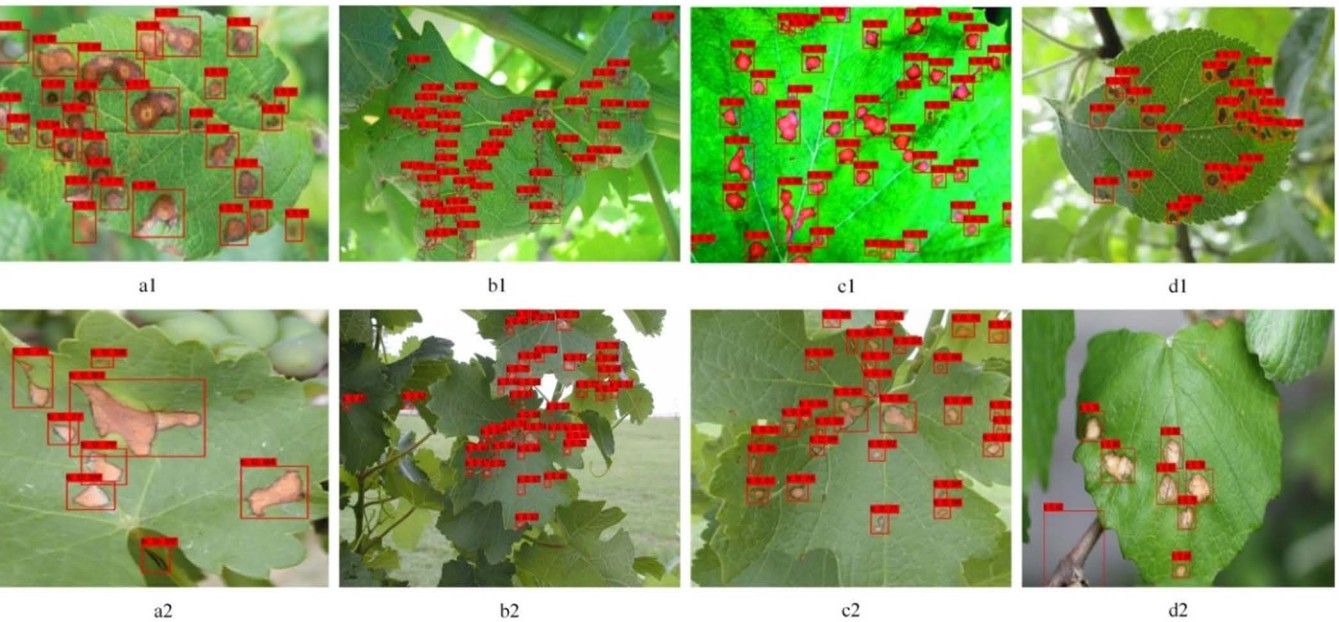

**Fig 12. Comprehensive outcomes of the Test2 dataset, showcasing the meticulous detection of various foliage disorders across distinct plant species. Column A presents the outcomes for single-bladed configurations; Column B displays the consequences of multi-bladed settings; Column C delineates the effects in the absence of obstacles; and Column D illustrates the impacts under obstructed conditions.**

as a disease. Similarly, background blurring was also applied to the images in Column b. Thus, despite the influence of obstructive backgrounds, the blurring of backgrounds during capture can reduce the error rate of detection. Comparing the detection results of Columns a and b, it is apparent that gaps caused by overlapping leaves were misidentified as lesions, thereby reducing the accuracy of lesion detection. Additionally, the mutual occlusion of leaves led to changes in lighting conditions, resulting in some lesions being missed. It can thus be concluded that the acquisition of single image data during capture will enhance the model's detection results. In subsequent research, the project team will delve deeper into plant diseases in natural environments, integrating considerations of imaging angles, lighting intensity, obstacle occlusion, and multi-leaf backgrounds during image acquisition. This will be one of the primary focuses of future research, aiming to continuously optimize the existing model and develop an efficient mobile detection system.

## Conclusion

For the disease detection of black rot on grape leaves, a YOLOv8-based disease detection model is constructed in this paper. In the improved network, SPD-Conv is used to replace each step roll-over layer and each pooling layer. It effectively solves the problem that the features of small targets are lost during the network training process. Meanwhile, the EMA module is added after each C2f in the Neck part of the network. It effectively improves the feature processing ability of the model for small targets and reduces the network computation. The experimental test results show that the improved YOLOv8 has a precision of 92.64%, a recall of 93.28%, and an AP of 96.17%. Compared with the mainstream networks YOLO v4, YOLO v5, YOLO v6, YOLO v7, and YOLOv8, the evaluation metrics are mentioned to be improved. Therefore, the improved network proposed in this paper can improve the detection accuracy of small target objects, as well as more accurately identify the black rot disease of grape leaves. n subsequent research endeavors, taking into account the shortcomings and issues identified in this paper, we shall delve deeper into the detection of plant diseases in natural environments. We aim to broaden the scope of detected objects, continually refine and enhance existing models, and provide technical support for the deployment on mobile devices.

## Author contributions

**Conceptualization:** Jinlin Qiu.

**Data curation:** Shouwen Chen.

**Methodology:** Shuxi Chen.

**Writing – original draft:** Jiajun Zhu.

**Writing – review & editing:** Haifei Zhang.

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
