## [Decision Letter · Decision Letter 0]

8 Jan 2025

PONE-D-24-43809An Application of YOLOv8 Integrated with Attention Mechanisms for Detection of Grape Leaf Black Rot SpotsPLOS ONE

Dear Dr. Zhang,

Thank you for submitting your manuscript to PLOS ONE. After careful consideration, we feel that it has merit but does not fully meet PLOS ONE’s publication criteria as it currently stands. Therefore, we invite you to submit a revised version of the manuscript that addresses the points raised during the review process.

We look forward to receiving your revised manuscript.

Kind regards,

Narendra Khatri, Ph.D.

Academic Editor

PLOS ONE

Journal requirements: When submitting your revision, we need you to address these additional requirements. 1. Please ensure that your manuscript meets PLOS ONE's style requirements, including those for file naming. The PLOS ONE style templates can be found at https://journals.plos.org/plosone/s/file?id=wjVg/PLOSOne_formatting_sample_main_body.pdf and https://journals.plos.org/plosone/s/file?id=ba62/PLOSOne_formatting_sample_title_authors_affiliations.pdf. 2. PLOS requires an ORCID iD for the corresponding author in Editorial Manager on papers submitted after December 6th, 2016. Please ensure that you have an ORCID iD and that it is validated in Editorial Manager. To do this, go to ‘Update my Information’ (in the upper left-hand corner of the main menu), and click on the Fetch/Validate link next to the ORCID field. This will take you to the ORCID site and allow you to create a new iD or authenticate a pre-existing iD in Editorial Manager. 3. Please note that PLOS ONE has specific guidelines on code sharing for submissions in which author-generated code underpins the findings in the manuscript. In these cases, we expect all author-generated code to be made available without restrictions upon publication of the work. Please review our guidelines at https://journals.plos.org/plosone/s/materials-and-software-sharing#loc-sharing-code and ensure that your code is shared in a way that follows best practice and facilitates reproducibility and reuse. 4. Thank you for stating the following financial disclosure:  [Natural Science Foundation of Nantong City (JC2023075); 2)Social Livelihood Science and Technology Plan of Nantong City (MSZ2023175). 3)Jiangsu Provincial Natural Science Foundation for Higher Education Institutions (22KJB520032); ,].  Please state what role the funders took in the study.  If the funders had no role, please state: ""The funders had no role in study design, data collection and analysis, decision to publish, or preparation of the manuscript."" If this statement is not correct you must amend it as needed. Please include this amended Role of Funder statement in your cover letter; we will change the online submission form on your behalf.

Additional Editor Comments:

Minor Revision

Reviewers' comments:

Reviewer's Responses to Questions

**Comments to the Author**

1. Is the manuscript technically sound, and do the data support the conclusions?

Reviewer #1: Yes

Reviewer #2: Yes

2. Has the statistical analysis been performed appropriately and rigorously? 

Reviewer #1: Yes

Reviewer #2: Yes

3. Have the authors made all data underlying the findings in their manuscript fully available?

Reviewer #1: Yes

Reviewer #2: Yes

4. Is the manuscript presented in an intelligible fashion and written in standard English?

Reviewer #1: Yes

Reviewer #2: Yes

5. Review Comments to the Author

Reviewer #1: 1. no furture work

2. what are the challenges faced

3. the work is well done and experimented, the imporved method result is okay

4. the gramatical errors was avoided

5. similaruty test shows that it is 10-11% which is okay

Reviewer #2: 1- The introduction did not define the work environment of the research field, nor did it address its general basics or the research gap it suffers from.

2- Add a paragraph at the end of the introduction that indicates the structure of the research with all its parts.

3- Refer to the dataset used with a reference that contains it and add it to the references.

4- Explaining the difference between CNN, and Spatial Pyramid Dilated Convolution

5- Clarify future work on the presented work and expand conclusions based on a well-constructed discussion of the numerical values of the results in tables and figures.

6- Include additional references published in the years 2023-2024

6. PLOS authors have the option to publish the peer review history of their article (what does this mean?). If published, this will include your full peer review and any attached files.

Reviewer #1: No

Reviewer #2: No

---

## [Author Response · Author response to Decision Letter 1]

20 Feb 2025

Reviewer #1

1.no furture work

Thank you for your suggestions. Based on your advice, we have added information about the direction of future research in the introduction section. We hope to receive your approval.

2.what are the challenges faced

Thank you for your suggestions. Based on your advice, we have added information about the direction of future research in the introduction section. We hope to receive your approval. We have expanded the content in Section 4 by adding two new subsections: 4.1 "Previous Research" and 4.5 "Analysis of Test Results in Natural Environments."

3.the work is well done and experimented, the imporved method result is okay

Thank you for your acknowledgment. We will continue to follow up on the subsequent research.

4. the gramatical errors was avoided

We have reviewed the entire text based on your suggestions. Additionally, we have invited a professional grammar teacher to further refine the manuscript. You will be able to see the improvements in our newly submitted version.

5. similaruty test shows that it is 10-11% which is okay

Thank you for your acknowledgment. We will continue to follow up on the subsequent research.

Reviewer #2

1- The introduction did not define the work environment of the research field, nor did it address its general basics or the research gap it suffers from.

Thank you very much for your valuable feedback. Based on your guidance, we have thoroughly revised the introduction section. We have included a detailed description of the main work of this paper, the current challenges encountered, and the future research directions of our research group. You will notice these additions in our newly submitted manuscript. We hope our modifications meet with your approval.

2- Add a paragraph at the end of the introduction that indicates the structure of the research with all its parts.

Thank you very much for your invaluable suggestions. Following your guidance, we have provided an explanation in the introduction section under "The main work of this paper is as follows." The focus of this paper lies in improving the algorithm while comparing it with the YOLO system algorithm. Additionally, we have expanded the content in Section 4 by incorporating two new subsections: 4.1 "Previous Research" and 4.5 "Analysis of Test Results in Natural Environments."

3- Refer to the dataset used with a reference that contains it and add it to the references.

Thank you for your suggestion. We have added the citation for the dataset in the References section. We hope this meets with your approval.

4- Explaining the difference between CNN, and Spatial Pyramid Dilated Convolution

Thank you for your inquiry. We have made extensive modifications in the section titled "Grape Leaf Spot Detection Based on Improved YOLOv8." In the newly submitted manuscript, we have provided a more detailed elaboration, which includes addressing your concerns and re-explaining the workflow and detailed steps of the network. We hope the improved manuscript will meet with your approval.

5- Clarify future work on the presented work and expand conclusions based on a well-constructed discussion of the numerical values of the results in tables and figures.

This issue certainly warrants our attention. Our research has consistently focused on disease detection. In the revised manuscript, we have included our previous related work titled "Grape Leaf Black Rot Detection Based on Super-Resolution Image Enhancement and Deep Learning." Additionally, we have added Section 4.5, "Analysis of Test Results in Natural Environments." To further evaluate the advantages of the improved network, we have introduced the "PT" metric. In Table 3, we have also conducted further discussions and analyses. We hope the enhanced manuscript will meet with your approval.

6- Include additional references published in the years 2023-2024

Thank you for your feedback. We have made revisions to the manuscript. As you will see in the newly submitted version, the citations in the introduction section have been reorganized. We have included recent related research and data analysis. We hope the revised manuscript meets with your approval.

---

## [Decision Letter · Decision Letter 1]

11 Mar 2025

An Application of YOLOv8 Integrated with Attention Mechanisms for Detection of Grape Leaf Black Rot Spots

PONE-D-24-43809R1

Dear Dr. Zhang,

We’re pleased to inform you that your manuscript has been judged scientifically suitable for publication and will be formally accepted for publication once it meets all outstanding technical requirements.

Kind regards,

Narendra Khatri, Ph.D.

Academic Editor

PLOS ONE

Additional Editor Comments (optional):

Accepted

Reviewers' comments:

Reviewer's Responses to Questions

**Comments to the Author**

1. If the authors have adequately addressed your comments raised in a previous round of review and you feel that this manuscript is now acceptable for publication, you may indicate that here to bypass the “Comments to the Author” section, enter your conflict of interest statement in the “Confidential to Editor” section, and submit your "Accept" recommendation.

Reviewer #1: All comments have been addressed

Reviewer #2: All comments have been addressed

2. Is the manuscript technically sound, and do the data support the conclusions?

Reviewer #1: Yes

Reviewer #2: (No Response)

3. Has the statistical analysis been performed appropriately and rigorously? 

Reviewer #1: Yes

Reviewer #2: (No Response)

4. Have the authors made all data underlying the findings in their manuscript fully available?

Reviewer #1: Yes

Reviewer #2: (No Response)

5. Is the manuscript presented in an intelligible fashion and written in standard English?

Reviewer #1: Yes

Reviewer #2: (No Response)

6. Review Comments to the Author

Reviewer #1: The authors should ensure that all figures are clearly labeled and described in the text.

The manuscript would benefit from a discussion on the potential for generalizing the proposed method to other plant diseases or small target detection tasks.

A brief discussion on ethical considerations would add depth to the manuscript.

Reviewer #2: (No Response)

7. PLOS authors have the option to publish the peer review history of their article (what does this mean?). If published, this will include your full peer review and any attached files.

Reviewer #1: No

Reviewer #2: No

---

## [Editor Report · Acceptance letter]

PONE-D-24-43809R1

PLOS ONE

Dear Dr. Zhang,

I'm pleased to inform you that your manuscript has been deemed suitable for publication in PLOS ONE. Congratulations! Your manuscript is now being handed over to our production team.

Kind regards,

on behalf of

Dr. Narendra Khatri

Academic Editor

PLOS ONE